**COMMENT**

# Psychological measures aren't toothbrushes

Malte Elson [1✉], Ian Hussey[1], Taym Alsalti[2] & Ruben C. Arslan [2]

Most psychological measures are used only once or twice. This proliferation and variability threaten the credibility of research. The Standardisation Of BEhavior Research (SOBER) guidelines aim to ensure that psychological measures are standardised and, unlike toothbrushes, reused by others.

Psychological constructs and measures suffer from the toothbrush problem[1]: no self-respecting psychologist wants to use anyone else's. This causes proliferation: many measures are used only once or twice (Fig. 1) and there is no tendency for researchers to agree on measures over time (Fig. 2). Proliferation happens because researchers promote their own brands, because discovering reusable measures in the large, fragmented academic literature is difficult, and simply for its own sake. At first glance, measurement proliferation may seem negligible, or even beneficial – after all, who would oppose studying the same phenomenon through multiple methodological approaches? Here, we argue that proliferation is in fact a serious barrier to cumulative science.

## A jingle-jangle of labels
Some measures actually quantify different things, but share similar labels (or even identical ones: In APA PsycTests, no less than 19 different tests go by "theory of planned behavior questionnaire", 15 by "job satisfaction scale", and 11 by "self-efficacy scale"). Other measures quantify the same thing as existing measures but under a different label. Known as the Jingle and Jangle fallacies, these are common and well-documented threats to the replicability and validity of psychological research, e.g. in studies on emotion[2]. They involve a nominal fallacy: that a measure's name tells you about its contents or what it measures[3].

## Undisclosed flexibility
Even when authors profess using the same measure of the same construct, all is not yet well because disclosed and undisclosed measurement flexibility, i.e. changes to a measure with known or unknown psychometric consequences, is common[4]. Dropping, adding, and altering items in self-report scales, aggregating total scores in various ways in laboratory tasks, or varying stimuli and trial durations all occur while researchers not only refer to the same construct, but actually to the same nominal instrument[5]. Even when all decisions are disclosed, only a methodological literature review will reveal that many studies used, for instance, unique aggregation algorithms, scoring strategies, or items, often with unknown psychometric consequences.

[1] Institute of Psychology, University of Bern, Fabrikstrasse 8, 3012 Bern, Switzerland. [2] Department of Psychology, University of Leipzig, Leipzig, Germany.
✉email: malte.elson@unibe.ch

**Fig. 1 Measurement proliferation in psychology.** Most psychological measures are used only once or twice, as coded in APA PsycInfo, which has records of tests and measures used in each study. Widely used tests are almost all from clinical psychology. Labelled tests were used more than 10,000 times: PHQ-9 patient health questionnaire-9, RSES Rosenberg self-esteem scale, HADS Hospital anxiety and depression scale, PANSS positive and negative syndrome scale, HAM-D Hamilton rating scale for depression, BDI Beck depression inventory, BDI-II Beck depression inventory-II. Further details in the SI.

### Generalising across samples

Current test norms are necessary for valid comparisons of individual test scores to population values[6]. Even widely used measures have typically never been normed in the population in which they are being used, or the available test norms are badly outdated. Without current norms, users of the measures cannot judge whether their sample selection procedure engendered bias, which makes it harder to judge generalisability. Supposedly 'standardised' effect sizes, such as correlations, are biased when all that is available for standardisation is the restricted within-sample standard deviation[7]. Such effect sizes may not be directly aggregated across samples without complex corrections - but nonetheless routinely are in meta-analyses.

Literatures that seem coherent and rigorous to the casual or even the experienced reader might in fact be anything but. For a reasonable synthesis of the evidence, meta-analysts would have to correct for the differences in sample selection, variability, reliability, and any other measurement-driven sources of heterogeneity. However, doing so properly is often difficult because the required information is missing[7] and bias correction techniques rely on often untestable assumptions.

Hence, (a) the lack of strong empirical or procedural norms in measurement, (b) the lack of transparency in reporting, and (c) the lack of common referents (i.e., test norms) in measurement are an enormous threat to meaningful evidence cumulation and research synthesis. For instance, to boost the reliability coefficient of an established scale in a primary study, researchers frequently drop what they argue to be 'poorly performing' items[8]. However, this approach is itself unreliable and produces inconsistent recommendations which item to drop[9]. Without also obtaining an out-of-sample norm for this modified measure, or a replication showing that dropping the chosen item(s) consistently

improves measurement, this does not improve the current or future use of the measurement-especially when such item dropping is not fully reported, as is often the case[8]: a measure cannot be improved when changes to it are not communicated.

Consequently, when measurement flexibility is present studies with measures using the same label may not be meaningfully compared, either directly with each other or in a larger research synthesis. Further, new validity evidence for an existing measure can only be applied quite narrowly, or with great uncertainty when it is unclear which studies match the validated protocol. And even the most peculiar decisions in a study can remain unnoticed, and their implications unknown, if no meaningful comparison against a proven standard can be made. Not being able to detect that study results are the outcome of a fishing expedition can result in a seemingly homogeneous literature that is actually the product of a trawling conglomerate.

### The SOBER guidelines

Psychology should be serious about standardising its measures - and it currently is not. But who should enforce this call to raise the bar on measurement and rein in ad-hockery? There are many stakeholders able to shape scientists' behaviours through meaningful policies, but we believe journals in general, and in particular psychology journals promoting robust science - including Communications Psychology, should implement policies to raise the quality of psychological science. Here, we propose the Standardisation Of BEhavior Research (SOBER) guidelines that specifically address issues of flexibility and norming in measurement (see Table 1).

### Moving forward

Across the psychological landscape, we call on research communities to (1) recognise measurement flexibility, similar to the widely acknowledged issues related to p-hacking, as a serious threat to scientific credibility, and (2) set and use continuously updated and validated standards for measurement and standard reporting guidelines that are maintained and updated with new evidence.

To this end, we envision an open repository of measurement protocols, one that fills the gap left by academic publishers after the decline of paper-pencil measures, with implementations for widely used study tools, test norms, standard scoring rules, and machine-readable metadata. This repository would facilitate discovery of measures, scrutiny of design decisions, meta-science through a systematic assessment of research practices. It would permit large language models to independently assess semantic overlap and reliability, and possibly even estimate discriminant and convergent validity, of newly developed measures against those already stored in a repository.

Scrutiny of the details of previous work's measures is necessary to both inform how we should interpret existing findings and to increase measures' future reuse potential. Transparency about the fine grain details of our measures allows others to reuse them with fidelity, and allows for the fidelity of measures to be checked between studies. These aspects of transparency and their scientific benefits have yet to be tapped by our field. If we want to build a cumulative evidence base in psychology, we need to standardise our measures and protocols. Psychologists need to stop remixing and recycling, and start reusing (measures, not toothbrushes).

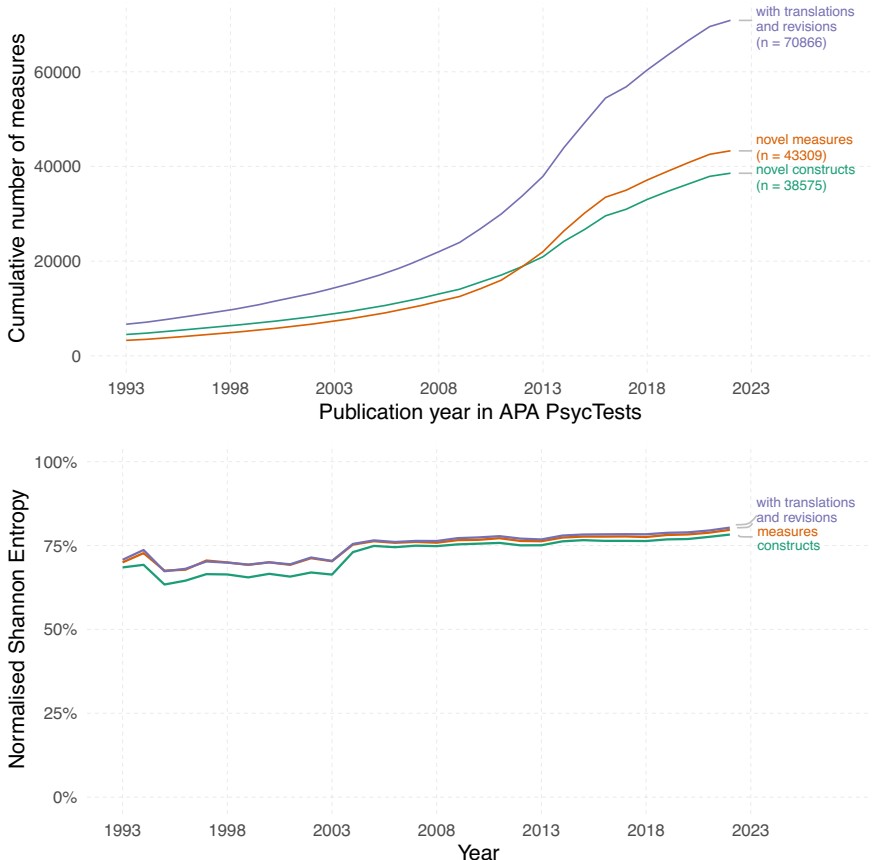

**Fig. 2 Fragmentation of the psychological literature.** The literature does not converge on standards over time, but rather remains fairly consistently fragmented. This trend is observed across novel constructs (green line), measures (orange line), and original measures lumped with their revisions and translations (purple line). Normalised Shannon entropy is a measure of heterogeneity, or fragmentation[10]. It takes varying numbers of publications per year into account and in our case has an intuitive meaning: closer to 0%, a few measures dominate, closer to 100%, measures are used an equal number of times across publications. Further details in the SI.

**Table 1 The SOBER policies for psychological journals; how authors comply with them, and how they should be enforced by editors and reviewers.**

| Policy | Compliance by authors | Enforcement by reviewers and editors |
|---|---|---|
| Demonstrate nonredundancy | When creating a new measure for a primary study, provide evidence of nonredundancy (with other measures and/or constructs) or incremental validity in a separate sample, e.g. through latent variable-based analyses of associations with a large selection of scales as opposed to simple correlations. | Require validation and norming in independent data, or an explanation why this is not necessary. Reject studies that use novel/ad-hoc measures without providing validity evidence from independent data. |
| Demonstrate protocol adherence | When following a measurement procedure published elsewhere, cite relevant protocols, and demonstrate you adhered to them (e.g., by sharing study materials). | Check claims of protocol adherence by comparing materials and cited sources. Journals dedicate resources to this task. |
| Justify Modifications | Prove that any deviation or modification to an existing measure is either meaningful (e.g. to address non-invariance of a measure between samples) or irrelevant, e.g. by providing validity evidence in a separate sample. Document when deviations happened, and if possible assess the robustness of conclusions. | Discourage authors from modifying existing scales by, for instance, dropping "poorly performing" items if the authors cannot show how these deviations affect the measure out-of-sample. Journals incentivise methodological research primarily providing validity evidence for commonly used measures rather than answering substantive research questions. |
| Preregistration and Registered Reports | Determine procedural and scoring details ahead of time, reporting all deviations as they potentially weaken the strength of conclusions. Analytical and statistical decisions should be preregistered with code rather than a narrative description. | Require authors to provide rationales for decision making in measurement specifically. Compare preregistrations (and any reported deviations) with what was actually employed and reported. |
| Comprehensive Reporting | Report all of the items, stimuli, instructions, procedural parameters or other measurement characteristics used in a study or generated during the development process. | Check for comprehensive data and materials beyond what is reported in the manuscript. |
| Facilitate Research Synthesis | Report standard deviations and means (regardless of whether data are shared), do not exclusively report effect sizes relative to the in-sample variation (so-called standardised effect sizes like Cohen's $d$, correlations, $R^2$). | Insist on complete descriptive statistics to make rigorous meta-analysis feasible. Suggest effect sizes be standardised by test norms instead of in-sample variation. |

For a more detailed version of this table, see Supplementary Table 1.

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

## Acknowledgements

This research was supported by the META-REP Priority Program of the German Research Foundation (#464488178). The funders had no role in the decision to publish or preparation of the manuscript. The authors thank the American Psychological Association for their support and providing access to the APA PsycTests database.

## Author contributions

Conceptualisation: M.E., I.H., T.A., R.C.A., visualisation: R.C.A., writing—original draft: M.E., and writing—review and editing: M.E., I.H., T.A., R.C.A.

## Competing interests

The authors declare no competing interests.
