## [Peer Review File · Communications Psychology]

20th Jul 23

Dear Malte,

Your Comment titled "Psychological Measures Aren't Toothbrushes" has now been seen by 2 referees, whose comments appear below. In the light of their advice I am delighted to say that we are happy, in principle, to publish it in Communications Psychology under a Creative Commons 'CC BY' open access license without charge.

We will not send your revised paper for further review if, in the editors' judgement, the referees' comments on the present version have been addressed. If the revised paper is in Communications Psychology format, in accessible style and of appropriate length, we shall accept it for publication immediately.

The reviewers raise some important points that will allow you to revise the Comment into a more persuasive and accessible piece. In some small instances, their requests run counter to our formatting guidelines (for example, we don't use keywords or abstracts in our Comments). I noted these issues on the attached edited version of your manuscript and in the checklist to avoid any confusion. I ask you to kindly attend to each comment in detail.

EDITORIAL REQUESTS:

* Please review the changes in the attached copy of your manuscript and address the comments and queries I have added. If using Word, please use the 'track changes' feature to make the process of accepting your manuscript more efficient. (We will require one marked-up and one "clean" version of your manuscript at resubmission).

* Please check whether your manuscript contains third-party images, such as figures from the literature, stock photos, clip art or commercial satellite and map data. If any of the display items in your manuscript (figures, tables, boxes or movies) include images that are the same as, or are adaptations of, previously published images, please fill in the ``Third Party Rights Table``, and return to us when you submit your revised manuscript. This information will enable us to obtain the necessary rights to re-use such material. If we are unable to obtain the necessary rights to use or adapt any of the material that you wish to use, we will contact you to discuss alternative options.

* Communications Psychology uses a transparent peer review system. On author request, confidential information and data can be removed from the published reviewer reports and rebuttal letters prior to publication. If you are concerned about the release of confidential data, please let us know specifically what information you would like to have removed. Please note that we cannot incorporate redactions for any other reasons.

*If you have not done so already, please alert me to any related manuscripts from your group that are under consideration or in press at other journals, or are being written up for submission to other journals (see www.nature.com/authors/editorial_policies/duplicate.html for details).

FORMATTING GUIDELINES:

You will find a complete list of formatting requirements following this link:

<https://www.nature.com/documents/commsj-style-formatting-checklist-comment.pdf>

Please use the checklist to prepare your manuscript for final submission. In the following, I also highlight some issues of particular importance.

** Title

Titles should be descriptive of the main message your manuscript conveys and should not exceed 90 characters (including spaces). Please note that punctuation is not allowed, nor are titles of the following format: "title: subtitle". Although the choice of title is largely yours, in light of the referees' feedback, we strongly recommend a change in title.

** Preface

The paper's preface (up to 40 words; without references) should serve both as a general introduction to the topic, and highlight your position or proposal. Because we hope that researchers across all fields of psychology will be interested in your work, the preface should be as accessible as possible. You may keep the present preface.

** Length

The ideal length for Review Article in Communications Psychology is 1,500 words. We have some flexibility, however, but please ensure that your text does not exceed 1,800 words.

** Main text

Please revise your section headings in the main text. These should relate to the content of the article rather than being generic or indecipherable. Headings should be no longer than 30 characters (including spaces) and should not use punctuation.

** Figures

Please remove all figures from the main text and upload them individually, one figure per file. To ensure the swift processing of your paper please provide the highest quality, vector format, versions of your images (.ai, .eps, .psd) where available. Text and labelling should be in a separate layer to enable editing during the production process. If vector files are not available then please supply the figures in whichever format they were compiled in and not saved as flat .jpeg or .TIFF files. If your artwork contains any photographic images, please ensure these are at least 300 dpi.

* Figures should be simple and informative — multi-part figures are best avoided.

* References

Please do not include more than 15 references. A Comment is an opinion piece and does not follow the same standards for referencing as research or review pieces. Widely known or easily explainable issues do not require a reference. Self-citations are explicitly discouraged. The purpose of references in a Comment is to point readers to literature that they may consider if they want to dive deeper

into a topic, not to support each individual statement in the piece.

References appear as superscript Arabic numerals, in order of mention. The reference list mentions references in the numerical order in which they are mentioned in the main text. If a reference is cited more than once, the same number is used throughout the text and the reference receives a single entry in the reference list.

Only papers that have been published or accepted by a named publication should be in the reference list (preprints and citations of datasets are also permitted). Unpublished/Submitted research should not be included in the reference list; it should only be mentioned briefly and parenthetically in the main text. Note that no major arguments should rely on unpublished research.

Published conference abstracts and URLs for web sites should be cited parenthetically in the text, not in the reference list.

Footnotes are not used.

* Competing interests

Please include a "Competing interests" statement after the References. Note that we ask authors to declare both financial and non-financial competing interests. For more details, see <https://www.nature.com/authors/policies/competing.html>. If you have no financial or non-financial competing interests, please state so: "The authors declare no competing interests."

SUBMISSION INFORMATION:

* If you wish, you may also submit a visually arresting image, together with a concise legend, for consideration as a 'Hero Image' on our homepage. The file should be 1400x400 pixels and should be uploaded as 'Related Manuscript File'. In addition to our home page, we may also use this image (with credit) in other journal-specific promotional material.

In order to accept your paper, we require the following:

* A cover letter describing your response to our editorial requests.

* A separate document detailing your point-by-point response to any issues raised by our referees (please include the referees' comments in this document).

* The final version of your text as a Word or TeX/LaTeX file, with any tables prepared using the Table menu in Word or the table environment in TeX/LaTeX and using the 'track changes' feature in Word.

* Production-quality versions of all figures, supplied as separate files. Photographic images should be 300 dpi in RGB format (.jpg, TIFF or native Photoshop format) and any labels/scale bars included in a separate layer from the image. Line art, graphs and schemes should be vector format (.ai, .eps, .pdf); Adobe Illustrator files are preferred and will minimize production time. Any chemical structures or schemes contained within figures should additionally be supplied as separate Chemdraw (.cdx) files.

At acceptance, the corresponding author will be required to complete an Open Access Licence to Publish on behalf of all authors, declare that all required third party permissions have been obtained.

Please note that your paper cannot be sent for typesetting to our production team until we have received this information; **therefore, please ensure that you have this ready when submitting the final version of your manuscript.**

ORCID

Communications Psychology is committed to improving transparency in authorship. As part of our efforts in this direction, we are now requesting that all authors identified as 'corresponding author' create and link their Open Researcher and Contributor Identifier (ORCID) with their account on the Manuscript Tracking System (MTS) prior to acceptance. ORCID helps the scientific community achieve unambiguous attribution of all scholarly contributions. For more information please visit <http://www.springernature.com/orcid>

For all corresponding authors listed on the manuscript, please follow the instructions in the link below to link your ORCID to your account on our MTS before submitting the final version of the manuscript. If you do not yet have an ORCID you will be able to create one in minutes.

IMPORTANT: All authors identified as 'corresponding author' on the manuscript must follow these instructions. Non-corresponding authors do not have to link their ORCIDs but are encouraged to do so. Please note that it will not be possible to add/modify ORCIDs at proof. Thus, if they wish to have their ORCID added to the paper they must also follow the above procedure prior to acceptance.

To support ORCID's aims, we only allow a single ORCID identifier to be attached to one account. If you have any issues attaching an ORCID identifier to your MTS account, please contact the Platform Support Helpdesk.

[link redacted]

We hope to hear from you within two weeks; please let us know if the process may take longer.

Best

Marike

Marike Schiffer, PhD

Chief Editor
Communications Psychology

REVIEWERS' COMMENTS:

Reviewer #1 (Remarks to the Author):

A timely and informative work on an important topic, which will be helpful and useful for a wide audience. There has been quite a lot of talk wrt jingle-jangle fallacies in psychological science recently, but this work certainly offers more, and from a different angle. Constructive comments towards amendment of what already is a convincing ms. are offered directly below.

Title: the paper's title is very concise, if not fragmentary (no cues as to what type of research and design it contains). I would reconsider adding a subtitle (including phrases such as „meta-research“, „SOBER guidelines“). Also, as is evident from the ref list (Mischel, 2008), the title is inspired by thoughts and ideas of Mischel (2008) and, as such, the title is an allusion (i.e., one would have to know Mischel's contribution, in order to understand the allusive title). Perhaps explain these circumstances a little bit, for the sake of comprehensibility, and credit Mischel more explicitly.

I did not find an abstract (just a „Preface“), or a keyword list. The Preface lacks a statement regarding the intention and aims of proposing the SOBER guidelines, or a kind of outlook.

The section titles (Jibber-Jabber, Jiggle-Joggle, General-Jibble, Journal-Jolly, Gist-Jab) are very short (see comment above wrt the paper's title), and these phrases do not really reappear in the main text, nor are they explained there. In particular, it appears as if there is no general ref to jingle-jangle fallacies (an emerging methodology research topic over the course of the past few years). I would suggest to add explanations, as international readers, non-native English speakers in all likelihood will have difficulties in understanding the respective meaning of phrases like „Gist-Jab“, „Journal-Jolly“ (try Google searches of these terms).

I did not find a figure note for the first figure („Times used in studies“, etc.). The necessary details and procedural info should be provided here (namely, how this was precisely done, etc.). In similar vein, for the 2 further figures. With regards to Fig. 3, the fragmentation measure (standardised entropy) needs brief explanation and a ref.

Main, text, end of „Jiggle-Joggle“ section, the passage with refs Elson (2019) and Steegen et al. (2016) is very condensed, e.g., reader not already familiar with multiverse-style approaches of data analysis won't grasp the meaning. Hence, some elaboration here would be beneficial.

End of first para of „General-Jibble“ section: many meta-analysts will object to the statement that standardized effect sizes „cannot be directly compared or combined across samples ... in meta-analyses“: think of psychometric meta-analysis (the Hunter-Schmidt approach), correction formulas (for range restriction, and so forth), or effect-moderator analysis in meta-analysis (i.e., coding for study and/or samples features). Some elaboration needed here, and a more cautious statement.

Similarly, the next but one para in the same section, namely the statement „common item selection methods are themselves unreliable and produce inconsistent recommendations which item to

drop“, etc.: what about automatic item-selection methods, such as those applied in nonparametric item response theory (Mokken scaling, AISP approach).

The SOBER guidelines are a great idea, I liked this much, but is there already a researcher-friendly checklist document (to fill out), apart from the criteria an overview provided in Table 1? It would also be interesting how the SOBER guidelines were created (e.g., internally developed in a group of collaborating researchers?, with external feedback, or without?).

RE Table 1: „demonstrate nonredundancy“ is akin to the incremental validity of a measure, a common textbook term in the domains of scale construction and test validation. Should be mentioned here as well. „Prereqs and RRs“: brief definitions, along with appropriate refs, lacking here. And (bottom line, central column, of this table): „report standard deviations and means“, this would, of course, be redundant with open data policies (suggestion: mention this).

Last but two para of main text, „large language models“, „semantic overlap“: it would be good to have refs also here.

Last para, last sentence, „... and start reusing“: I would suggest to also add „corroborating“ and „refining“ here.

Reviewer #2 (Remarks to the Author):

Review Report

Psychological Measures Aren't Toothbrushes

Date of review: 17 July 2023

Comments:

1. I applaud the authors to choose a topic in the areas of replication and psychological measurement. In my view, this connection is much-needed and has the potential to bring to attention the quality of psychological measures.
2. I am afraid, I was stumbling over the title already. I was not able to make a connection to the topics discussed in the paper. I suggest making it more precise. The same applies to the headings, except for the jingle-jangle fallacies, which are actual terms in the APA dictionary. I guess that the authors intended to make the paper a “funny” and “light” read with these headings. However, this did not work for me—I was confused about the imprecise headings, and they actually made the paper harder to read. I suggest making them more precise, too.
3. Figure 2: As far as I understand these figures, they aim to show frequencies as a function of “year”. First, which year is meant here, study year, publication year etc. I suggest specifying this. Second, given that the authors presumably only have data for a given year, the times in between years do not have data points. Hence, in my view, the graph should rather be a histogram instead of a plot with lines.
4. What exactly is “measurement flexibility”. I am afraid, I could not find an explicit definition in the paper.
5. Did the authors generate the data underlying Figures 1 and 2 themselves? I could not trace back the sources or the material documenting this. For the sake of openness, please provide the details

for this.

6. One of the recommendations the authors make is that we need to “standardize our measures and protocols” (p. 7). While I generally agree, I find these claim a bit too short-sighted, because standardizing measures ultimately brings up the question of what the population of standardization will be. Example: Intelligence tests have been standardized with a limited population that does not allow any generalization to some specific groups of people. Moreover, psychological measures are sometimes not fully invariant across groups of samples (e.g., self-concept measures across cultures or countries) and may thus not be “standardized” at all. I am wondering what the authors think about these lines of thinking.

I hope that these comments will somehow be helpful for the authors’ work.

Dear Marike,

Dear reviewers,

First of all, we appreciate all the effort put in reading and considering the manuscript. We are particularly grateful for all the valuable suggestions and advice provided in your comments. Thus, we did our best to implement all the improvements suggestions and remarks in the revised version of the manuscript.

In the following, we have grouped your comments into thematic blocks and address them accordingly. All changes to the manuscript are tracked.

***R#1:** A timely and informative work on an important topic, which will be helpful and useful for a wide audience. There has been quite a lot of talk wrt jingle-jangle fallacies in psychological science recently, but this work certainly offers more, and from a different angle. Constructive comments towards amendment of what already is a convincing ms. are offered directly below.*

***R#2:** I applaud the authors to choose a topic in the areas of replication and psychological measurement. In my view, this connection is much-needed and has the potential to bring to attention the quality of psychological measures.*

We thank both reviewers for their kind and encouraging words.

Title, Subheadings, Keywords, and ‘Hero Image’

***Editor:** Titles should be descriptive of the main message your manuscript conveys and should not exceed 90 characters (including spaces). Please note that punctuation is not allowed, nor are titles of the following format: "title: subtitle". Although the choice of title is largely yours, in light of the referees' feedback, we strongly recommend a change in title.*

***R#1:** Title: the paper's title is very concise, if not fragmentary (no cues as to what type of research and design it contains). I would reconsider adding a subtitle (including phrases such as „meta-research“, „SOBER guidelines“). Also, as is evident from the ref list (Mischel, 2008), the title is inspired by thoughts and ideas of Mischel (2008) and, as such, the title is an allusion (i.e., one would have to know Mischel's contribution, in order to understand the allusive title). Perhas explain these circumstances a little bit, for the sake of comprehensibility, and credit Mischel more explicitly.*

***R#2:** I am afraid, I was stumbling over the title already. I was not able to make a connection to the topics discussed in the paper. I suggest making it more precise.*

Although this was more a stylistic rather than a substantive request, it was probably the one we most struggled with (as you can probably tell from the email I sent you about this). The reasons are that we really like the title, and we also believe that it will make readers more curious, not less, about the paper. We wanted to give this some serious consideration, so we came up with a few more

descriptive alternatives and showed these, and the original, to several of our colleagues -- all suggested we use the original. Now, we are not oblivious to the possibilities that our colleagues are either very like-minded, or that they simply being nice to us. At the risk of being wrong here, we have decided to retain it, and hope that you agree with this decision. We also have rephrased the introduction to credit Mischel more explicitly, alleviating some of the confusion that the reviewers have mentioned.

Editor: *Please revise your section headings in the main text. These should relate to the content of the article rather than being generic or indecipherable. Headings should be no longer than 30 characters (including spaces) and should not use punctuation.*

R#1: *The section titles (Jibber-Jabber, Jiggle-Joggle, General-Jibble, Journal-Jolly, Gist-Jab) are very short (see comment above wrt the paper's title), and these phrases do not really reappear in the main text, nor are they explained there. In particular, it appears as if there is no general ref to jingle-jangle fallacies (an emerging methodology research topic over the course of the past few years). I would suggest to add explanations, as international readers, non-native English speakers in all likelihood will have difficulties in understanding the respective meaning of phrases like „Gist-Jab“, „Journal-Jolly“ (try Google searches of these terms).*

R#2: *The same applies to the headings, except for the jingle-jangle fallacies, which are actual terms in the APA dictionary. I guess that the authors intended to make the paper a “funny” and “light” read with these headings. However, this did not work for me—I was confused about the imprecise headings, and they actually made the paper harder to read. I suggest making them more precise, too.*

The subheadings have been revised to be more descriptive.

Editor: The paper's preface (up to 40 words; without references) should serve both as a general introduction to the topic, and highlight your position or proposal. Because we hope that researchers across all fields of psychology will be interested in your work, the preface should be as accessible as possible. You may keep the present preface.

R#1: *I did not find an abstract (just a „Preface“), or a keyword list. The Preface lacks a statement regarding the intention and aims of proposing the SOBER guidelines, or a kind of outlook.*

The editor has kindly pointed us to Nature's guidelines for comment-type articles, which state that we should not include keywords. Given the constraints of the preface length, we also have decided to keep the submitted version.

Editor: *If you wish, you may also submit a visually arresting image, together with a concise legend, for consideration as a 'Hero Image' on our homepage. The file should be 1400x400*

pixels and should be uploaded as 'Related Manuscript File'. In addition to our home page, we may also use this image (with credit) in other journal-specific promotional material.

We have submitted a Hero Image as instructed.

Methodological Details for Figures 1 and 2

R#1: *I did not find a figure note for the first figure („Times used in studies“, etc.). The necessary details and procedural info should be provided here (namely, how this was precisely done, etc.). In similar vein, for the 2 further figures. With regards to Fig. 3, the fragmentation measure (standardised entropy) needs brief explanation and a ref.*

R#2: *3. Figure 2: As far as I understand these figures, they aim to show frequencies as a function of “year”. First, which year is meant here, study year, publication year etc. I suggest specifying this.*

R#2: *Second, given that the authors presumably only have data for a given year, the times in between years do not have data points. Hence, in my view, the graph should rather be a histogram instead of a plot with lines.*

R#2: *Did the authors generate the data underlying Figures 1 and 2 themselves? I could not trace back the sources or the material documenting this. For the sake of openness, please provide the details for this.*

We have revised the figures and their labels and captions for clarity. We have created a more detailed description of the data and methodology underlying our Figures 1 and 2 in a Supplementary Information file. Finally, we choose to retain the line graph instead of a histogram; we believe it is absolutely standard practice to plot publication-related information by year this way. We were not sure what R#2 meant by “times in between years” as the data we report are for consecutive years between 1993 and 2023.

Various Minor Clarifications

R#1: *Main, text, end of „Jiggle-Joggle“ section, the passage with refs Elson (2019) and Steegen et al. (2016) is very condensed, e.g., reader not already familiar with multiverse-style approaches of data analysis won't grasp the meaning. Hence, some elaboration here would be beneficial.*

This section has been revised, during which we also removed the Steegen et al. reference as this was not central the point we were making here.

R#1: *End of first para of „General-Jibble“ section: many meta-analysts will object to the statement that standardized effect sizes „cannot be directly compared or combined across samples ... in meta-analyses“: think of psychometric meta-analysis (the Hunter-Schmidt approach), correction formulas (for range restriction, and so forth), or effect-moderator*

analysis in meta-analysis (i.e., coding for study and/or samples features). Some elaboration needed here, and a more cautious statement.

We agree with the reviewer, and have revised the wording here to be more cautious.

R#1: *Similarly, the next but one para in the same section, namely the statement „common item selection methods are themselves unreliable and produce inconsistent recommendations which item to drop“, etc.: what about automatic item-selection methods, such as those applied in nonparametric item response theory (Mokken scaling, AISP approach).*

Again, the reviewer makes a very good point. We have revised the wording here to refer specifically to reliability threshold-based item deletion. Although we share the reviewer's intuition that automatic selection methods should yield more reliable deletion recommendations, we are not aware of another paper specifically presenting evidence for this idea that we could cite. A comprehensive discussion of the difference between selection methods is beyond the scope of this paper, however.

R#1: *„Preregs and RRs“: brief definitions, along with appropriate refs, lacking here.*

The maximum number of references permitted is only 15, and we believe that, particularly for the readership of Communications Psychology, preregistration and registered reports can be assumed as shared knowledge. Ultimately, we would refer this question to the editor.

R#1: *Last but two para of main text, „large language models“, „semantic overlap“: it would be good to have refs also here.*

For reasons of space, we chose not to include any references for this rather broad suggestion we were making.

R#1: *Last para, last sentence, „... and start reusing“: I would suggest to also add „corroborating“ and „refining“ here.*

We have retained this sentence as it was originally, but made changes to the paper, particularly the SOBER table, to highlight that we absolutely encourage principled refinement of measures, where appropriate. We think from the context it is clear that the final sentence in this paragraph only concerns the use of redundant measures with unknown psychometric properties.

R#2: *4. What exactly is “measurement flexibility”. I am afraid, I could not find an explicit definition in the paper.*

We now provide a definition of measurement flexibility, and thank the reviewer for this suggestion.

***R#2:** One of the recommendations the authors make is that we need to “standardize our measures and protocols” (p. 7). While I generally agree, I find these claim a bit too short-sighted, because standardizing measures ultimately brings up the question of what the population of standardization will be. Example: Intelligence tests have been standardized with a limited population that does not allow any generalization to some specific groups of people. Moreover, psychological measures are sometimes not fully invariant across groups of samples (e.g., self-concept measures across cultures or countries) and may thus not be “standardized” at all. I am wondering what the authors think about these lines of thinking.*

The reviewer raises an important point that is central to the core of our argument, and the reason we wrote this paper. It is exactly those principled modifications that the reviewer describes (e.g., adapting a test to suit a new population) that we are strongly in favor of – what we are concerned with is the proliferation of redundant measures and unprincipled measurement flexibility with unknown psychometric consequences. It is this research practice that we believe needs to be addressed with increasing standardisation (and demands of journals to use standardised instruments where possible).

The SOBER Guidelines Table

***R#1:** RE Table 1: „demonstrate nonredundancy“ is akin to the incremental validity of a measure, a common textbook term in the domains of scale construction and test validation. Should be mentioned here as well.*

We have included this point in the first row of the table.

***R#1:** And (bottom line, central column, of this table): „report standard deviations and means“, this would, of course, be redundant with open data policies (suggestion: mention this).*

Although the reviewer is correct that such values could be computed from publicly available data, there may still be reasons to consistently report them: 1) researchers may commit critical errors when handling an unknown, potentially complex dataset; 2) authors of papers may have committed an error in their analyses or preparation of a shareable data file, and a reporting of a value in the paper that does not match those extracted from shared data could be of value; 3) and, lastly, simply for convenience for readers and, potentially, meta-analysts extracting such values.

***R#1:** The SOBER guidelines are a great idea, I liked this much, but is there already a researcher-friendly checklist document (to fill out), apart from the criteria an overview provided in Table 1? It would also be interesting how the SOBER guidelines were created (e.g.,*

internally developed in a group of collaborating researchers?, with external feedback, or without?).

We appreciate the idea of a usable document for the SOBER criteria. Upon drafting this document, however, we felt that a checklist covering different types of research (RCTs, surveys, etc.) and methodologies (self-report, behavioral assessment, psychophysiology, etc) would generate a lot of checklist items that only applicable in particular cases. Therefore, we have created a table that instead offers broader questions as guidance for reviewers to address flexibility and standardisation issues. This table is submitted as Supplementary Information.